# WSNet: Compact and Efficient Networks with Weight Sampling

## Abstract

We present a new approach and a novel architecture, termed WSNet, for learning compact and efficient deep neural networks. Existing approaches conventionally learn full model parameters independently and then compress them via *ad hoc* processing such as model pruning or filter factorization. Alternatively, WSNet proposes learning model parameters by sampling from a compact set of learnable parameters, which naturally enforces parameter sharing throughout the learning process. We demonstrate that such a novel weight sampling approach (and induced WSNet) promotes both weights and computation sharing favorably. By employing this method, we can more efficiently learn much smaller networks with competitive performance compared to baseline networks with equal numbers of convolution filters. Specifically, we consider learning compact and efficient 1D convolutional neural networks for audio classification. Extensive experiments on multiple audio classification datasets verify the effectiveness of WSNet. Combined with weight quantization, the resulted models are up to $\mathbf{180\times}$ smaller and theoretically up to $\mathbf{16\times}$ faster than the well-established baselines, without noticeable performance drop.

## 1 Introduction

Despite remarkable successes in various applications, including *e.g.* audio classification, speech recognition and natural language processing, deep neural networks (DNNs) usually suffer following two problems that stem from their inherent huge parameter space. First, most of state-of-the-art deep architectures are prone to over-fitting even when trained on large datasets (Simonyan & Zisserman, 2015; Szegedy et al., 2015). Secondly, DNNs usually consume large amount of storage memory and energy (Han et al., 2016). Therefore these networks are difficult to embed into devices with limited memory and power (such as portable devices or chips). Most existing networks aim to reduce computational budget through network pruning (Han et al., 2015; Anwar et al., 2017; Li et al., 2017; Collins & Kohli, 2014), filter factorization (Jaderberg et al., 2014; Lebedev et al., 2014), low bit representation (Rastegari et al., 2016) for weights and knowledge transfering (Hinton et al., 2015). In contrast to the above works that ignore the strong dependencies among weights and learn filters independently based on existing network architectures, this paper proposes to explicitly enforce the parameter sharing among filters to more effectively learn compact and efficient deep networks.

In this paper, we propose a **W**eight **S**ampling deep neural network (*i.e.* WSNet) to significantly reduce both the model size and computation cost of deep networks, achieving more than $100\times$ smaller size and up to $16\times$ speedup at negligible performance drop or even achieving better performance than the baseline (*i.e.* conventional networks that learn filters independently). Specifically, WSNet is parameterized by layer-wise *condensed filters* from which each filter participating in actual convolutions can be directly sampled, in both spatial and channel dimensions. Since condensed filters have significantly fewer parameters than independently trained filters as in conventional CNNs, learning by sampling from them makes WSNet a more compact model compared to conventional CNNs. In addition, to reduce the ubiquitous computational redundancy in convolving the overlapped filters and input patches, we propose an integral image based method to dramatically reduce the computation cost of WSNet in both training and inference. The integral image method is also advantageous because it enables weight sampling with different filter size and minimizes computational overhead to enhance the learning capability of WSNet.

In order to demonstrate the efficacy of WSNet, we conduct extensive experiments on the challenging acoustic scene classification and music detection tasks. On each test dataset, including MusicDet200K (a self-collected dataset, as detailed in Section 4), ESC-50 (Piczak, 2015a), Urban-Sound8K (Salamon et al., 2014) and DCASE (Stowell et al., 2015a), WSNet significantly reduces the model size of the baseline by 100× with comparable or even higher classification accuracy. When compressing more than 180×, WSNet is only subject to negligible accuracy drop. At the same time, WSNet significantly reduces the computation cost (up to 16×). Such results strongly establish the capability of WSNet to learn compact and efficient networks. Although we detailed experiments mostly limited to 1D CNNs in this paper, we will explore how the same approach can be naturally generalized to 2D CNNs in future work.

## 2 RELATED WORKS

### 2.1 AUDIO CLASSIFICATION

In this paper we considered Acoustic Scene Classification (ASC) tasks as well as music detection tasks. ASC aims to classify the surrounding environment where an audio stream is generated given the audio input (Barchiesi et al., 2015). It can be applied in many different ways such as audio tagging (Cai et al., 2006), audio collections management (Landone et al., 2007), robotic navigation (Chu et al., 2006), intelligent wearable interfaces (Xu et al., 2008), context adaptive tasks (Schilit et al., 1994), etc. Music detection is a related task to determine whether or not a small segment of audio is music. It is usually treated as a binary classification problem given an audio segment as input, *i.e.*, to classify the segment into two categories: music or non-music.

As evident in many other areas, convolutional neural networks (CNN) have been widely used in audio classification tasks (Valenti et al., 2016) (Salamon & Bello, 2017). SoundNet (Aytar et al., 2016) stands out among different CNNs for sound classification due to the following two reasons. First, it is trained from the large-scale unlabeled sound data using visual information as a bridge, while many other networks are trained with smaller datasets. Secondly, SoundNet directly takes one dimensional raw wave signals as input so that there is no need to calculate time-consuming audio specific features, *e.g.* MFCC (Pols et al., 1966) (Davis & Mermelstein, 1980) and spectrogram (Flanagan, 1972). SoundNet has yielded significant performance improvements on state-of-the-art results with standard benchmarks for acoustic scene classification. In this paper, we demonstrate that the proposed WSNet achieves a comparable or even better performance than SoundNet at a significantly smaller size and faster speed.

### 2.2 DEEP MODEL COMPRESSION AND ACCELERATION

Early approaches for deep model compression include (LeCun et al., 1989; Hassibi & Stork, 1993) that prune the connections in networks based on the second order information. Most recent works in network compression adopt weight pruning (Han et al., 2015; Collins & Kohli, 2014; Anwar et al., 2017; Lebedev & Lempitsky, 2016; Kim et al., 2015; Luo et al., 2017; Li et al., 2017), filter decomposition (Sindhwani et al., 2015; Denton et al., 2014; Jaderberg et al., 2014), hashed networks (Chen et al., 2015; 2016) and weight quantization (Han et al., 2016). However, although those works reduce model size, they also suffer from large performance drop. Bucilu et al. (2006) and Ba & Caruana (2014) are based on student-teacher approches which may be difficult to apply in new tasks since they require training a teacher network in advance. Denil et al. (2013) predicts parameters based on a few number of weight values. Jin et al. (2016) proposes an iterative hard thresholding method, but only achieve relatively small compression ratios. Gong et al. (2014) uses a binning method which can only be applied over fully connected layers. Hinton et al. (2015) compresses deep models by transferring the knowledge from pre-trained larger networks to smaller networks. In contrast, WS-Net is able to learn compact representation for both convolution layers and fully connected layers from scratch. The deep models learned by WSNet can significantly reduce model size compared to the baselines with comparable or even better performance.

In terms of deep model acceleration, the factorization and quantization methods listed above can also reduce computation latency in inference. While irregular pruning (as done in most pruning methods (Han et al., 2016)) incurs computational overhead, grouped pruning (Lebedev & Lempitsky, 2016) is able to accelerate networks. FFT (Mathieu et al., 2013) and LCNN (Bagherinezhad et al.,

2016) are also used to speed up computation in pratice. Comparatively, WSNet is superior because it learns networks that have both smaller model size and faster computation versus baselines.

## 2.3 EFFICIENT MODEL DESIGN

WSNet presents a class of novel models with the appealing properties of a small model size and small computation cost. Some recently proposed efficient model architectures include the class of Inception models (Szegedy et al., 2015; Ioffe & Szegedy, 2015; Chollet, 2016) which adopts depth-wise separable convolutions, the class of Residual models (He et al., 2016; Xie et al., 2017; Chen et al., 2017) which uses residual path for efficient optimization, and the factorized networks which use fully factorized convolutions. MobileNet (Howard et al., 2017) and Flattened networks (Jin et al., 2014) are based on factorization convolutions. ShuffleNet (Zhang et al., 2017) uses group convolution and channel shuffle to reduce computational cost. Compared with above works, WS-Net presents a new model design strategy which is more flexible and generalizable: the parameters in deep networks can be obtained conveniently from a more compact representation, *e.g.* through the weight sampling method proposed in this paper or other more complex methods based on the learned statistic models.

## 3 METHOD

In this section, we describe details of the proposed WSNet for 1D CNNs. First, the notations are introduced. Secondly, we elaborate on the core components in WSNet: *weight sampling* along the spatial dimension and channel dimension. Thirdly, we introduce the denser weight sampling to enhance the learning capability of WSNet. Finally, we propose an integral image method for accelerating WSNet in both training and inference.

### 3.1 NOTATIONS

Before diving into the details, we first introduce the notations used in this paper. The traditional 1D convolution layer takes as input the feature map $\mathbf{F} \in \mathbb{R}^{T \times M}$ and produces an output feature map $\mathbf{G} \in \mathbb{R}^{T \times N}$ where $(T, M, N)$ denotes the spatial length of input, the channel of input and the number of filters respectively. We assume that the output has the same spatial size as input which holds true by using zero padded convolution. The 1D convolution kernel $\mathbf{K}$ used in the actual convolution of WSNet has the shape of $(L, M, N)$ where $L$ is the kernel size. Let $\mathbf{k}_n, n \in \{1, \cdots N\}$ denotes a filter and $\mathbf{f}_t, t \in \{1, \cdots T\}$ denotes a input patch that spatially spans from $t$ to $t + L - 1$, then the convolution assuming stride one and zero padding is computed as:

$$\mathbf{G}_{t,n} = \mathbf{f}_t \cdot \mathbf{k}_n = \sum_{l=0}^{L-1} \sum_{m=0}^{M-1} \mathbf{F}_{t+l,m} \times \mathbf{K}_{l,m,n}, \tag{1}$$

where $\cdot$ stands for the vector inner product. Note we omit the element-wise activation function to simplify the notation.

In WSNet, instead of learning each weight independently, $\mathbf{K}$ is obtained by sampling from a learned *condensed filter* $\mathbf{\Phi}$ which has the shape of $(L^*, M^*)$. The goal of training WSNet is thus cast to learn more compact DNNs which satisfy the condition of $L^*M^* < LMN$. To quantize the advantage of WSNet in achieving compact networks, we define the *compactness* of $\mathbf{K}$ in a learned layer in WSNet w.r.t. the conventional layer with independently learned weights as:

$$\text{compactness} = \frac{LMN}{L^*M^*}.$$

In the following section, we demonstrate WSNet learn compact networks by sampling weights in two dimensions: the spatial dimension and the channel dimension.

### 3.2 WEIGHT SAMPLING

#### 3.2.1 ALONG SPATIAL DIMENSION

In conventional CNNs, the filters in a layer are learned independently which presents two disadvantages. Firstly, the resulted DNNs have a large number of parameters, which impedes their deploy-

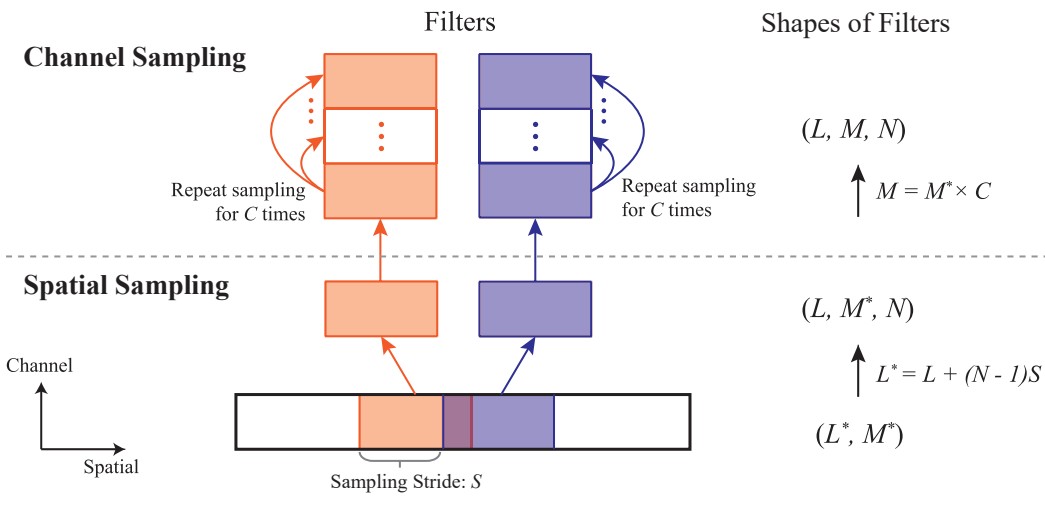

Figure 1: Illustration of WSNet that learns small condensed filters with weight sampling along two dimensions: spatial dimension (the bottom panel) and channel dimension (the top panel). The figure depicts procedure of generating two continuous filters (in pink and purple respectively) that convolve with input. In **spatial sampling**, filters are extracted from the condensed filter with a stride of $S$. In **channel sampling**, the channel of each filter is sampled repeatedly for $C$ times to achieve equal with the input channel. Please refer to Section 3.2 for detailed explanations.

ment in computation resource constrained platforms. Second, such over-parameterization makes the network prone to overfitting and getting stuck in (extra introduced) local minimums. To solve these two problems, a novel weight sampling method is proposed to efficiently reuse the weights among filters. Specifically, in each convolutional layer of WSNet, all convolutional filters $\mathbf{K}$ are sampled from the condensed filter $\mathbf{\Phi}$, as illustrated in Figure 1. By scanning the weight sharing filter with a window size of $L$ and stride of $S$, we could sample out $N$ filters with filter size of $L$. Formally, the equation between the filter size of the condensed filter and the sampled filters is:

$$L^* = L + (N - 1)S. \tag{2}$$

The *compactness* along spatial dimension is $\frac{LM^*N}{L^*M^*} \approx \frac{L}{S}$. Note that since the minimal value of $S$ is 1, the minimal value of $L^*$ (*i.e.* the minimum spatial length of the condensed filter) is $L + N - 1$ and the maximal achievable compactness is therefore $L$.

### 3.2.2 ALONG CHANNEL DIMENSION

Although it is experimentally verified that the weight sampling strategy could learn compact deep models with negligible loss of classification accuracy (see Section 4), the maximal compactness is limited by the filter size $L$, as mentioned in Section 3.2.1.

In order to seek more compact networks without such limitation, we propose a channel sharing strategy for WSNet to learn by weight sampling along the channel dimension. As illustrated in Figure 1 (top panel), the actual filter used in convolution is generated by repeating sampling for $C$ times. The relation between the channels of filters before and after channel sampling is:

$$M = M^* \times C, \tag{3}$$

Therefore, the *compactness* of WSNet along the channel dimension achieves $C$. As introduced later in Experiments (Section 4), we observe that the repeated weight sampling along the channel dimension significantly reduces the model size of WSNet without significant performance drop. One notable advantage of channel sharing is that the maximum compactness can be as large as $M$ (*i.e.* when the condensed filter has channel of 1), which paves the way for learning much more aggressively smaller models (*e.g.* more than 100× smaller models than baselines).

The above analysis for weight sampling along spatial/channel dimensions can be conveniently generalized from convolution layers to fully connected layers. For a fully connected layer, we treat its weights as a flattened vector with channel of 1, along which the spatial sampling (ref. Section 3.2.1) is performed to reduce the size of learnable parameters. For example, for the fully connected layer "fc1" in the baseline network in Table 1, its filter size, channel number and filter number are 1536, 1 and 256 respectively. We can therefore perform spatial sampling for "fc1" to learn a more compact representation. Compared with convolutional layers which generally have small filter sizes and thus have limited compactnesses along the spatial dimenstion, the fully connected layers can achieve larger compactnesses along the spatial dimension without harming the performance, as demonstrated in experimental results (ref. to Section 4.2).

### 3.2.3 THE TRAINING OF CONDENSED FILTERS

WSNet is trained from the scratch in a similar way to conventional deep convolutional networks by using standard error back-propagation. Since every weight $\mathbf{K}_{l,m,n}$ in the convolutional kernel $\mathbf{K}$ is sampled from the condensed filter $\mathbf{\Phi}$ along the spatial and channel dimension, the only difference is the gradient of $\mathbf{\Phi}_{i,j}$ is the summation of all gradients of weights that are tied to it. Therefore, by simply recording the position mapping $\mathcal{M} : (i, j) \to (l, m, n)$ from $\mathbf{\Phi}_{i,j}$ to all the tied weights in $\mathbf{K}$, the gradient of $\mathbf{\Phi}_{i,j}$ is calculated as:

$$\frac{\partial \mathcal{L}}{\partial \mathbf{\Phi}_{i,j}} = \sum_{s \in \mathcal{M}(i,j)} \frac{\partial \mathcal{L}}{\partial \mathbf{K}_s} \tag{4}$$

where $\mathcal{L}$ is the conventional cross-entropy loss function. In open-sourced machine learning libraries which represent computation as graphs, such as TensorFlow Abadi et al. (2016), Equation (4) can be calculated automatically.

## 3.3 DENSER WEIGHT SAMPLING

The performance of WSNet might be adversely affected when the size of condensed filter is decreased aggressively (*i.e.* when $S$ and $C$ are large). To enhance the learning capability of WSNet, we could sample more filters for layers with significantly reduced sizes. Specifically, we use a smaller sampling stride $\bar{S}$ ($\bar{S} < S$) when performing spatial sampling. In order to keep the shape of weights unchanged in the following layer, we append a $1 \times 1$ convolution layer with the shape of $(1, \bar{n}, n)$ to reduce the channels of densely sampled filters. It is experimentally verified that denser weight sampling can effectively improve the performance of WSNet in Section 4. However, since it also brings extra parameters and computational cost to WSNet, denser weight sampling is only used in lower layers of WSNet whose filter number ($n$) is small. Besides, one can also conduct channel sampling on the added $1 \times 1$ convolution layers to further reduce their sizes.

## 3.4 EFFICIENT COMPUTATION WITH INTEGRAL IMAGE

According to Equation 1, the computation cost in terms of the number of multiplications and adds (*i.e.* Mult-Adds) in a conventional convolutional layer is:

$$TMLN \tag{5}$$

However, as illustrated in Figure 2, since all filters in a layer in WSNet are sampled from a condensed filter $\mathbf{\Phi}$ with stride $S$, calculating the results of convolution in the conventional way as in Eq. (1) incurs severe computational redundance. Concretely, as can be seen from Eq. (1), one item in the ouput feature map is equal to the summation of $L$ inner products between the row vector of $\mathbf{f}$ and the column vector of $\mathbf{k}$. Therefore, when two overlapped filters that are sampled from the condensed filter (*e.g.* $\mathbf{k}_1$ and $\mathbf{k}_2$ in Fig. 2) convolves with the overlapped input windows (*e.g.* $\mathbf{f}_1$ and $\mathbf{f}_2$ in Fig. 2)), some partially repeated calculations exist (*e.g.* the calculations highlight in green and indicated by arrow in Fig. 2). To eliminate such redundancy in convolution and speed-up WSNet, we propose a novel integral image method to enable efficient computation via sharing computations.

We first calculate an inner product map $\mathbf{P} \in \mathbb{R}^{T \times L^*}$ which stores the inner products between each row vector in the input feature map (*i.e.* $\mathbf{F}$) and each column vector in the condensed filter (*i.e.* $\mathbf{\Phi}$):

$$\mathbf{P}(u, v) = \begin{cases} \mathbf{F}_{u,:} \cdot \mathbf{\Phi}_{:,v}, & u \in [0, T-1] \text{ and } v \in [0, L^*-1] \\ 0, & otherwise. \end{cases} \tag{6}$$

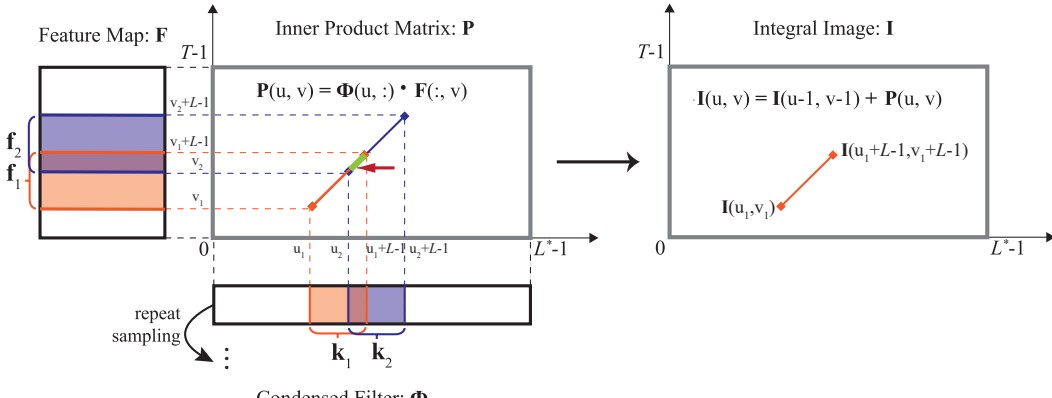

Figure 2: Illustration of efficient computation with integral image in WSNet. The inner product map $\mathbf{P} \in \mathbb{R}^{T \times L^*}$ calculates the inner product of each row in $\mathbf{F}$ and each column in $\mathbf{\Phi}$ as in Eq. (6). The convolution result between a filter $\mathbf{k}_1$ which is sampled from $\mathbf{\Phi}$ and the input patch $\mathbf{f}_1$ is then the summation of all values in the segment between $(u, v)$ and $(u + L - 1, v + L - 1)$ in $\mathbf{P}$ (recall that $L$ is the convolutional filter size). Since there are repeated calculations when the filter and input patch are overlapped, e.g. the green segment indicated by arrow when performing convolution between $\mathbf{k}_2$ and $\mathbf{s}_2$, we construct the integral image $\mathbf{I}$ using $\mathbf{P}$ according to Eq. (7). Based on $\mathbf{I}$, the convolutional results between any sampled filter and input patch can be retrieved directly in time complexity of $O(1)$ according to Eq. (8), e.g. the results of $\mathbf{k}_1 \cdot \mathbf{s}_1$ is $\mathbf{I}(u_1 + L - 1, v_1 + L - 1) - \mathbf{I}(u_1 - 1, v_1 - 1)$. For notation definitions, please refer to Sec. 3.1. The comparisons of computation costs between WSNet and the baselines using conventional architectures are introduced in Section 3.4.

The integral image for speeding-up convolution is denoted as $\mathbf{I}$. It has the same size as $\mathbf{P}$ and can be conveniently obtained throught below formulation:

$$\mathbf{I}(u, v) = \begin{cases} \mathbf{I}(u - 1, v - 1) + \mathbf{P}(u, v), & u > 0, v > 0 \\ \mathbf{P}(u, 0), & v = 0 \\ \mathbf{P}(0, v), & u = 0 \end{cases} \tag{7}$$

Based on $\mathbf{I}$, all convolutional results can be obtained in time complexity of $O(1)$ as follows

$$\mathbf{G}_{t,n} = \mathbf{I}(t + L - 1, nS + L - 1) - \mathbf{I}(t - 1, nS - 1) \tag{8}$$

Recall that the $n$-th filter lies in the spatial range of $(nS, nS + L - 1)$ in the condensed filter $\mathbf{\Phi}$. Since $\mathbf{G} \in \mathbb{R}^{T \times N}$, it thus takes $TN$ times of calculating Eq. (8) to get $\mathbf{G}$. In Eq. (6) $\sim$ Eq. (8), we omit the case of padding for clear description. When zero padding is applied, we can freely get the convolutional results for the padded areas even without using Eq. (8) since $\mathbf{I}(u, v) = \mathbf{I}(T, v-1), u > T$.

Based on Eq. (6) $\sim$ Eq. (8), the computation cost of the proposed integral image method is

$$\underbrace{TML^*}_{\text{Eq. (6)}} + \underbrace{TL^*}_{\text{Eq. (7)}} + \underbrace{TN}_{\text{Eq. (8)}} = T(M + 1)L^* + TN. \tag{9}$$

Note the computation cost of $\mathbf{P}$ (i.e. Eq. (6)) is the dominating term in Eq. (9). Based on Eq. (5), Eq. (9) and Eq. (2), the theoretical acceleration ratio is

$$\frac{TMLN}{T(M + 1)L^* + TN} \approx \frac{L}{S}$$

Recall that $L$ is the filter size and $S$ is the pre-defined stride when sampling filters from the condensed filter $\mathbf{\Phi}$ (ref. to Eq. (2)).

In practice, we adopt a variant of above method to further boost the computation efficiency of WSNet, as illustrated in Fig 3. In Eq. (6), we repeat $\mathbf{\Phi}$ by $C$ times along the channel dimension to

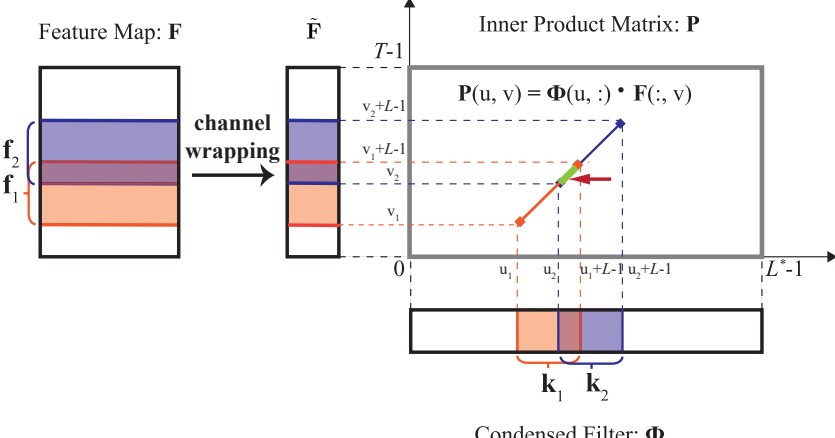

Figure 3: A variant of the integral image method used in practice which is more efficient than that illustrated in Figure 2. Instead of repeatedly sampling along the channel dimension of $\mathbf{\Phi}$ to convolve with the input $\mathbf{F}$, we wrap the channels of $\mathbf{F}$ by summing up $C$ matrixes that are evenly divided from $\mathbf{F}$ along the channels, *i.e.* $\tilde{\mathbf{F}}(i,j) = \sum_{c=0}^{C-1} \mathbf{F}(i, j + cM^*)$. Since the channle of $\tilde{\mathbf{F}}$ is only $1/C$ of the channel of $\mathbf{F}$, the overall computation cost is reduced as demonstrated in Eq. (10).

make it equal with the channel of the input $\mathbf{F}$. However, we could first wrap the channels of $\mathbf{F}$ by accumulating the values with interval of $L$ along its channel dimension to a thinner feature map $\tilde{\mathbf{F}} \in \mathbb{R}^{T \times M^*}$ which has the same channel number as $\mathbf{\Phi}$, *i.e.* $\tilde{\mathbf{F}}(i,j) = \sum_{c=0}^{C-1} \mathbf{F}(i, j + cM^*)$. Both Eq. (7) and Eq. (8) remain the same. Then the computational cost is reduced to

$$\underbrace{TM^*(C-1)}_{\text{channel warp}} + \underbrace{TM^*L^*}_{\text{Eq. (6)}} + \underbrace{TL^*}_{\text{Eq. (7)}} + \underbrace{TN}_{\text{Eq. (8)}} \tag{10}$$

where the first item is the computational cost of warping the channels of $\mathbf{F}$ to obtain $\tilde{\mathbf{F}}$. Since the dominating term (*i.e.* Eq. (6)) in Eq (10) is smaller than in Eq. (9), the overall computation cost is thus largely reduced. By combining Eq. (10) and Eq. (5), the theoretical acceleration compared to the baseline is

$$\frac{TMLN}{TM^*(C + L^* - 1) + T(L^* + N)} \tag{11}$$

Finally, we note that the integral image method applied in WSNet naturally takes advantage of the property in weight sampling: redundant computations exist between overlapped filters and input patches. Different from other deep model speedup methods (Sindhwani et al., 2015; Denton et al., 2014) which require to solve time-consuming optimization problems and incur performance drop, the integral image method can be seamlessly embedded in WSNet without negatively affecting the final performance.

## 4 EXPERIMENTS

In this section, we present the details and analysis of the results in our experiments. Extensive ablation studies are conducted to verify the effectiveness of the proposed WSNet on learning compact and efficient networks. On all tested datasets, WSNet is able to improve the classification performance over the baseline networks while using $\mathbf{100\times}$ smaller models. When using even smaller (*e.g.* $\mathbf{180\times}$) model size, WSNet achieves comparable performance w.r.t the baselines. In addition, WSNet achieves $2\times \sim 4\times$ acceleration compared to the baselines with a much smaller model (more than $100\times$ smaller).

## 4.1 Experimental Settings

**Datasets** We collect a large-scale music detection dataset (MusicDet200K) from publicly available platforms (*e.g.* Facebook, Twitter, *etc.*) for conducting experiments. For fair comparison with previous literatures, we also test WSNet on three standard, publicly available datasets, *i.e* ESC-50, UrbanSound8K and DCASE. The details of used datasets are as follows.

*MusicDet200K* aims to assign a sample a binary label to indicate whether it is music or not. MusicDet200K has overall 238,000 annotated sound clips. Each has a time duration of 4 seconds and is resampled to 16000 Hz and normalized (Piczak, 2015b). Among all samples, we use 200,000/20,000/18,000 as train/val/test set. The samples belonging to "non-music" count for 70% of all samples, which means if we trivially assign all samples to be "non-music", the classification accuracy is 70%.

*ESC-50* (Piczak, 2015a) is a collection of 2000 short (5 seconds) environmental recordings comprising 50 equally balanced classes of sound events in 5 major groups (*animals*, *natural soundscapes and water sounds*, *human non-speech sounds*, *interior/domestic sounds* and *exterior/urban noises*) divided into 5 folds for cross-validation. Following Aytar et al. (2016), we extract 10 sound clips from each recording with length of 1 second and time step of 0.5 second (*i.e.* two neighboring clips have 0.5 seconds overlapped). Therefore, in each cross-validation, the number of training samples is 16000. In testing, we average over ten clips of each recording for the final classification result.

*UrbanSound8K* (Salamon et al., 2014) is a collection of 8732 short (around 4 seconds) recordings of various urban sound sources (*air conditioner*, *car horn*, *playing children*, *dog bark*, *drilling*, *engine idling*, *gun shot*, *jackhammer*, *siren* and *street music*). As in ESC-50, we extract 8 clips with the time length of 1 second and time step of 0.5 second from each recording. For those that are less than 1 second, we pad them with zeros and repeat for 8 times (*i.e.* time step is 0.5 second).

*DCASE* (Stowell et al., 2015a) is used in the Detection and Classification of Acoustic Scenes and Events Challenge (DCASE). It contains 10 acoustic scene categories, 10 training examples per category and 100 testing examples. Each sample is a 30-second audio recording. During training, we evenly extract 12 sound clips with time length of 5 seconds and time step of 2.5 seconds from each recording.

**Evaluation criteria** To demonstrate that WSNet is capable of learning more compact and efficient models than conventional CNNs, three evaluation criteria are used in our experiments: model size, the number of multiply and adds in calculation (mult-adds) and classification accuracy.

**Baseline networks** To test the scability of WSNet to different network architectures (*e.g.* whether having fully connected layers or not), two baseline networks are used in comparision. The baseline network used on MusicDet200K consists of 7 convolutional layers and 2 fully connected layers, using which we demonstrate the effectiveness of WSNet on both convolutional layers and fully connected layers. For fair comparison with previous literatures, we firstly modify the state-of-the-art SoundNet (Aytar et al., 2016) by applying pooling layers to all but the last convolutional layer. As can be seen in Table 5, this modification significantly boosts the performance of original SoundNet. We then use the modified SoundNet as a baseline on all three public datasets. The architectures of the two baseline networks are shown in Table 1 and Table 2 respectively.

**Weight Quantization** Similar to other works (Han et al., 2016; Rastegari et al., 2016), we apply weight quantization to further reduce the size of WSNet. Specifically, the weights in each layer are linearly quantized to $q$ bins where $q$ is a pre-defined number. By setting all weights in the same bin to the same value, we only need to store a small index of the shared weight for each weight. The size of each bin is calculated as $(\max(\mathbf{\Phi}) - \min(\mathbf{\Phi}))/q$. Given $q$ bins, we only need $\log_2(q)$ bits to encode the index. Assuming each weight in WSNet is represented using 4 bytes float number (32 bits) without weight quantization, the ratio of each layer's size before and after weight quantization is $\frac{32L^*M^*}{L^*M^*\log_2(q)+32q}$. Recall that $L^*$ and $M^*$ are the spatial size and the channel number of condensed filter. Since the condition $L^*M^* \gg q$ generally holds in most layers of WSNet, weight quantization is able to reduce the model size by a factor of $\frac{32}{\log_2(q)}$. Different from (Han et al., 2016; Rastegari et al., 2016) which learns the quantization during training, we apply weight quantization to WSNet

Table 1: Baseline-1: configurations of the baseline network used on MusicDet200K. Each convolutional layer is followed by a nonlinearity layer (*i.e.* ReLU), batch normalization layer and pooling layer, which are omitted in the table for brevity. The strides of all pooling layers are 2. The padding strategies adopted for both convolutional layers and fully connected layers are all "size preserving".

| Layer | conv1 | conv2 | conv3 | conv4 | conv5 | conv6 | conv7 | fc1 | fc2 |
|---|---|---|---|---|---|---|---|---|---|
| Filter sizes | 32 | 32 | 16 | 8 | 8 | 8 | 4 | 1536 | 256 |
| #Filters | 32 | 64 | 128 | 128 | 256 | 512 | 512 | 256 | 128 |
| Stride | 2 | 2 | 2 | 2 | 2 | 2 | 2 | 1 | 1 |
| #Params | 1K | 65K | 130K | 130K | 260K | 1M | 1M | 390K | 33K |
| Params (%) | 0.03 | 2.1 | 4.2 | 4.2 | 8.4 | 33.7 | 33.7 | 12.6 | 1.1 |
| #Mult-Adds ($10^8$) | 4.1 | 65.5 | 32.7 | 8.2 | 4.1 | 4.2 | 1.0 | 0.1 | 0.007 |
| Mult-Adds (%) | 3.4 | 54.5 | 27.3 | 6.8 | 3.4 | 3.5 | 0.9 | 0.1 | 0.005 |

Table 2: Baseline-2: configuration of the baseline network used on ESC-50, UrbanSound8K and DCASE. This baseline is adapted from SoundNet (Aytar et al., 2016) as detailed in Section 4.1. For brevity, the nonlinearity layer (*i.e.* ReLU), batch normalization layer and pooling layer following each convolutional layer are omitted. The kernel sizes for pooling layers following conv1-conv4 and conv5-conv7 are 8 and 4 respectively. The stride of every pooling layers is 2.

| Layer | conv1 | conv2 | conv3 | conv4 | conv5 | conv6 | conv7 | conv8 |
|---|---|---|---|---|---|---|---|---|
| Filter sizes | 64 | 32 | 16 | 8 | 4 | 4 | 4 | 8 |
| #Filters | 16 | 32 | 64 | 128 | 256 | 512 | 1024 | 1401 |
| Stride | 2 | 2 | 2 | 2 | 2 | 2 | 2 | 2 |
| #Params | 1K | 16K | 32K | 65K | 130K | 520K | 2M | 11M |
| Params (%) | 0.01 | 0.11 | 0.22 | 0.45 | 0.90 | 3.63 | 14.55 | 79.61 |
| #Mult-Adds ($10^8$) | 2.3 | 9.0 | 4.5 | 2.3 | 1.2 | 1.2 | 1.2 | 2.3 |
| Mult-Adds (%) | 9.4 | 37.7 | 18.8 | 9.5 | 4.8 | 4.8 | 5.3 | 9.6 |

after its training. In the experiments, we find that such an off-line way is sufficient to reduce model size without losing accuracy.

**Implementation details**  WSNet is implemented and trained from scratch in Tensorflow (Abadi et al., 2016). Following Aytar et al. (2016), the Adam (Kingma & Ba, 2014) optimizer, a fixed learning rate of 0.001, and momentum term of 0.9 and batch size of 64 are used throughout experiments. We initialized all the weights to zero mean gaussian noise with a standard deviation of 0.01. In the network used on MusicDet200K, the dropout ratio for the dropout layers (Srivastava et al., 2014) after each fully connected layer is set to be 0.8. The overall training takes 100,000 iterations.

## 4.2   RESULTS AND ANALYSIS

### 4.2.1   MUSICDET200K

**Ablation analysis**  Through controlled experiments, we investigate the effects of each component in WSNet on the model size, computational cost and classification accuracy. The comparative study results of different settings of WSNet are listed in Table 3. For clear description, we name WSNets with different settings by the combination of symbols S/C/SC[†]/D/Q. Please refer to the caption of Table 3 for detailed meanings.

*(1) Spatial sampling.* We test the performance of WSNet by using different sampling stride $S$ in spatial sampling. As listed in Table 3, $S_2$ and $S_4$ slightly outperforms the classification accuracy of the baseline, possibly due to reducing the overfitting of models. When the sampling stride is 8, *i.e.* the compactness in spatial dimension is 8 (ref. to Section 3.2.1), the classification accuracy of $S_8$ only drops slightly by 0.6%. Note that the maximum compactness along the spatial dimension is equal to the filter size, thus for the layer "conv7" which has a filter size of 4, its compactness is limited by 4 (highlighted by underline in Table 3) in $S_8$. Above results clearly demonstrate that the

Table 3: Ablative study of the effects of different settings of WSNet on the model size, computation cost (in terms of #mult-adds) and classification accuracy on MusicDet200K. For clear description, we name WSNets with different settings by the combination of symbols $S/C/SC^\dagger/D/Q$. "S" denotes the weight sampling along spatial dimension; "C" denotes the weight sampling along the channel dimension. "$SC^\dagger$" denotes the weight sampling of fully connected layers whose parameters can be seen as flattened vectors with channel of 1. "D" denotes denser filter sampling. "Q" denotes weight quantization. With a symbol occurred in the name, the corresponding component is used in WSNet. The numbers in subscripts of $S/C/SC^\dagger/D/Q$ denotes the maximum compactness (ref. to Sec. 3.1 for the definition of compactness) on spatial/channel dimension in all layers, the ratio of the number of filters in WSNet versus in the baseline, the compactness of fully connected layers and the ratio of WSNet's size before and after weight quantization, respectively. To avoid confusion, $SC^\dagger$ only occured in the names when both spatial and channel sampling are applied for convolutional layers. The model size and the computational cost are provided for the baseline. For the model size and #mult-adds of WSNet, we provide the ratio of the baseline's model size versus WSNet's model size and the ratio of the baseline's #Mult-Adds versus WSNet's #Mult-Adds.

| WSNet's settings | conv{1-3} | | | conv4 | | | conv5 | | | conv6 | | | conv7 | | | fc1/2 | Acc. | Model size | Mult-Adds |
| | S | C | A | S | C | A | S | C | A | S | C | A | S | C | A | $SC^\dagger$ | | | |
|---|---|---|---|---|---|---|---|---|---|---|---|---|---|---|---|---|---|---|---|
| Baseline | 1 | 1 | 1 | 1 | 1 | 1 | 1 | 1 | 1 | 1 | 1 | 1 | 1 | 1 | 1 | 1 | $88.9 \pm 0.1$ | 3M ($1\times$) | 1.2e10 ($1\times$) |
| $Baseline Q_4$ | 1 | 1 | 1 | 1 | 1 | 1 | 1 | 1 | 1 | 1 | 1 | 1 | 1 | 1 | 1 | 1 | $88.8 \pm 0.1$ | $4\times$ | $1\times$ |
| $S_2$ | 2 | 1 | 1 | 2 | 1 | 1 | 2 | 1 | 1 | 2 | 1 | 1 | 2 | 1 | 1 | 2 | $89.0 \pm 0.0$ | $2\times$ | $1\times$ |
| $S_4$ | 4 | 1 | 1 | 4 | 1 | 1 | 4 | 1 | 1 | 4 | 1 | 1 | 4 | 1 | 1 | 4 | $89.0 \pm 0.0$ | $4\times$ | $1.8\times$ |
| $S_8$ | 8 | 1 | 1 | 8 | 1 | 1 | 8 | 1 | 1 | 8 | 1 | 1 | $\underline{4}$ | 1 | 1 | 8 | $88.3 \pm 0.1$ | $5.7\times$ | $3.4\times$ |
| $C_2$ | 1 | 2 | 1 | 1 | 2 | 1 | 1 | 2 | 1 | 1 | 2 | 1 | 1 | 2 | 1 | 2 | $89.1 \pm 0.2$ | $2\times$ | $1\times$ |
| $C_4$ | 1 | 4 | 1 | 1 | 4 | 1 | 1 | 4 | 1 | 1 | 4 | 1 | 1 | 4 | 1 | 4 | $88.7 \pm 0.1$ | $4\times$ | $1.4\times$ |
| $C_8$ | 1 | 8 | 1 | 1 | 8 | 1 | 1 | 8 | 1 | 1 | 8 | 1 | 1 | 8 | 1 | 8 | $88.6 \pm 0.1$ | $8\times$ | $2.4\times$ |
| $S_4 C_4 SC_4^\dagger$ | 4 | 4 | 1 | 4 | 4 | 1 | 4 | 4 | 1 | 4 | 4 | 1 | 4 | 4 | 1 | 4 | $88.7 \pm 0.0$ | $11.1\times$ | $5.7\times$ |
| $S_8 C_8 SC_8^\dagger$ | 8 | 8 | 1 | 8 | 8 | 1 | 8 | 8 | 1 | 8 | 8 | 1 | 4 | 8 | 1 | 8 | $88.4 \pm 0.0$ | $23\times$ | **$16.4\times$** |
| $S_8 C_8 SC_8^\dagger D_2$ | 8 | 8 | 2 | 8 | 8 | 1 | 8 | 8 | 1 | 8 | 8 | 1 | 4 | 8 | 1 | 8 | **$89.2 \pm 0.1$** | $20\times$ | $3.8\times$ |
| $S_8 C_8 SC_{15}^\dagger D_2$ | 8 | 8 | 2 | 8 | 8 | 1 | 8 | 8 | 1 | 8 | 8 | 1 | 8 | 8 | 1 | 15 | $88.6 \pm 0.0$ | $42\times$ | $3.8\times$ |
| $S_8 C_8 SC_8^\dagger Q_4$ | 8 | 8 | 1 | 8 | 8 | 1 | 8 | 8 | 1 | 8 | 8 | 1 | 4 | 8 | 1 | 8 | $88.4 \pm 0.0$ | $92\times$ | **$16.4\times$** |
| $S_8 C_8 SC_{15}^\dagger D_2 Q_4$ | 8 | 8 | 2 | 8 | 8 | 1 | 8 | 8 | 1 | 8 | 8 | 1 | 8 | 8 | 1 | 15 | $88.5 \pm 0.1$ | **$168\times$** | $3.8\times$ |

spatial sampling enables WSNet to learn significantly smaller model with comparable accuracies w.r.t. the baseline.

*(2) Channel sampling.* Three different compactness along the channel dimension, *i.e.* 2, 4 and 8 are tested by comparing with baslines. It can be observed from Table 3 that $C_2$ and $C_4$ and $C_8$ have linearly reduced model size without incurring noticeable drop of accuracy. In fact, $C_2$ can even improve the accuracy upon the baseline, demonstrating the effectiveness of channel sampling in WSNet. When learning more compact models, $C_8$ demonstrates better performance compared to $S_8$ tha has the same compactness in the spatial dimension, which suggests we should focus on the channel sampling when the compactness along the spatial dimension is high.

We then simultaneously perform weight sampling on both the spatial and channel dimensions. As demonstrated by the results of $S_4 C_4 SC_4^\dagger$ and $S_8 C_8 SC_8^\dagger$, WSNet can learn highly compact models (more than $20\times$ smaller than baselines) without noticeable performance drop (less than 0.5%).

*(3) Denser weight sampling.* Denser weight sampling is used to enhance the learning capability of WSNet with aggressive compactness (*i.e.* when $S$ and $C$ are large) and make up the performance loss caused by sharing too much parameters among filters. As shown in Table 3, by sampling $2\times$ more filters in conv1, conv2 and conv3, $S_8 C_8 SC_8^\dagger D_2$ significantly outperforms the $S_8 C_8 SC_8^\dagger$. Above results demonstrate the effectiveness of denser weight sampling to boost the performance of WSNet.

*(4) Integral image for efficient computation.* As evidenced in the last column in Table 3, the proposed integral image method consistently reduces the computation cost of WSNet. For $S_8 C_8 SC_8^\dagger$ which is $23\times$ smaller than the baseline, the computation cost (in terms of #mult-adds) is significantly reduced by 16.4 times. Due to the extra computation cost brought by the $1\times1$ convolution in denser

Table 4: The configurations of the WSNet used on ESC-50, UrbanSound8K and DCASE. Please refer to Table 3 for the meaning of symbols S/C/D. Since the input lengths for the baseline are different in each dataset, we only provide the #Mult-Adds for UrbanSound8K. Note that since we use the ratio of baseline's #Mult-Adds versus WSNet's #Mult-Adds for one WSNet, the numbers corresponding to WSNets in the column of #Mult-Adds are the same for all dataset.

| WSNet's settings | conv{1-4} | | | conv5 | | | conv6 | | | conv7 | | | conv8 | | | Model size | Mult-Adds |
|---|---|---|---|---|---|---|---|---|---|---|---|---|---|---|---|---|---|
| | S | C | A | S | C | A | S | C | A | S | C | A | S | C | A | | |
| Baseline | 1 | 1 | 1 | 1 | 1 | 1 | 1 | 1 | 1 | 1 | 1 | 1 | 1 | 1 | 1 | 13M ($1\times$) | 2.4e9 ($1\times$) |
| $S_8C_4D_2$ | 4 | 4 | 2 | 4 | 4 | 1 | 4 | 4 | 1 | 4 | 4 | 1 | 8 | 4 | 1 | $25\times$ | $2.3\times$ |
| $S_8C_8D_2$ | 4 | 4 | 2 | 4 | 4 | 1 | 4 | 4 | 1 | 4 | 8 | 1 | 8 | 8 | 1 | $45\times$ | $2.4\times$ |

Table 5: Comparison with state-of-the-arts on ESC-50. All results of WSNet are obtained by 10-folder validation. Please refer to Table 3 for the meaning of symbols S/C/D/Q. The baseline used here is a simple modification of SoundNet with 8 convolution layers (refer to Section 4.1 for details), thus they have the same model size.

| Model | Settings | Acc. (%) | Model size |
|---|---|---|---|
| baseline | scratch init.; provided data | $66.0 \pm 0.25$ | 13M ($1\times$) |
| baselineQ$_4$ | scratch init.; provided data | $65.8 \pm 0.25$ | $4\times$ |
| WSNet | $S_8C_4D_2$ | $66.5 \pm 0.10$ | $25\times$ |
| WSNet | $S_8C_4D_2Q_4$ | $\mathbf{66.25 \pm 0.25}$ | $\mathbf{100\times}$ |
| WSNet | $S_8C_8D_2$ | $66.1 \pm 0.15$ | $45\times$ |
| WSNet | $S_8C_8D_2Q_4$ | $\mathbf{65.8 \pm 0.25}$ | $\mathbf{180\times}$ |
| Piczak ConvNet (Piczak, 2015b) | scratch init.; provided data | 64.5 | 28M |
| SoundNet (Aytar et al., 2016) | scratch init.; provided data | 51.1 | 13M |
| SoundNet (Aytar et al., 2016) | pre-training; extra data | 72.9 | 13M |

filter sampling, $S_8C_8SC_8^\dagger D_2$ achieves lower acceleration ($3.8\times$). Group convolution (Xie et al., 2017) can be used to alleviate the computation cost of the added $1\times1$ convolution layers. We will explore this direction in our future work.

*(5) Weight quantization.* It can be observed from Table 3 that by using 256 bins to represent each weight by one byte (*i.e.* 8bits), $S_8C_8SC_{15}^\dagger A_2Q_4$ is reduced to 1/168 of the baseline's model size while incurring only 0.1% accuracy loss. The above result demonstrates that the weight quantization is complementary to WSNet and they can be used jointly to effectively reduce the model size of WSNet. Since we do not use weight quantization to accelerate models in this paper, the WSNets before and after weight quantization have the same computational cost.

### 4.2.2 ESC-50

The comparison of WSNet with other state-of-the-arts on ESC-50 is listed in Table 5. The settings of WSNet used on ESC-50, UrbanSound8K and DCASE are listed in Table 4. Compared with the baseline, WSNet is able to significantly reduce the model size of the baseline by 25 times and 45 times, while at the same time improving the accuracy of the baseline by 0.5% and 0.1% respectively. The computation costs of WSNet are listed in Table 4, from which one can observe that WSNet achieves higher computational efficiency by reducing the #Mult-Adds of the baseline by $2.3\times$ and $2.4\times$, respectively. Such promising results again demonstrate the effectiveness of WS-Net on learning compact and efficient networks. After applying weight quantization to WSNet, its model size is reduced to only 1/180 of the baseline while the accuracy only slightly drops by 0.2%. Compared with the SoundNet trained from scratch with provided data, WSNets significantly outperform its classification accuracy by over 10% with more than $100\times$ smaller models. Using a transfer learning approach, SoundNet (Aytar et al., 2016) that is pre-trained using a large number of unlabeled videos achieves better accuracy than WSNet. However, since the training method is

Table 6: Comparison with state-of-the-arts on UrbanSound8K. All results of WSNet are obtained by 5-folder validation. Please refer to Table 3 for the meaning of symbols S/C/D/Q.

| Model | Settings | Acc. (%) | Model size |
|-------|----------|----------|------------|
| baseline | scratch init.; provided data | $70.39 \pm 0.31$ | 13M ($1\times$) |
| baseline$Q_4$ | scratch init.; provided data | $70.10 \pm 0.31$ | $4\times$ |
| WSNet | $S_8C_4D_2$ | $70.76 \pm 0.15$ | $25\times$ |
| WSNet | $S_8C_4D_2Q_4$ | $\mathbf{70.61 \pm 0.20}$ | $\mathbf{100\times}$ |
| WSNet | $S_8C_8D_2$ | $70.14 \pm 0.23$ | $45\times$ |
| WSNet | $S_8C_8D_2Q_4$ | $\mathbf{70.03 \pm 0.11}$ | $\mathbf{180\times}$ |
| Piczak ConvNet (Piczak, 2015b) | pre-computed features | 73.1 | 28M |

Table 7: Comparison with state-of-the-arts on DCASE. Note there are only 100 samples in testing set. Please refer to Table 3 for the meaning of symbols S/C/D/Q.

| Model | Settings | Acc. (%) | Model size |
|-------|----------|----------|------------|
| baseline | scratch init.; provided data | $85 \pm 0$ | 13M ($1\times$) |
| baseline$Q_4$ | scratch init.; provided data | $84 \pm 0$ | $4\times$ |
| WSNet | $S_8C_4D_2$ | $86 \pm 0$ | $25\times$ |
| WSNet | $S_8C_4D_2Q_4$ | $\mathbf{86 \pm 0}$ | $\mathbf{100\times}$ |
| WSNet | $S_8C_8D_2$ | $84 \pm 0$ | $45\times$ |
| WSNet | $S_8C_8D_2Q_4$ | $\mathbf{84 \pm 0}$ | $\mathbf{180\times}$ |
| RNH (Roma et al., 2013) | scratch init.; provided data | 77 | - |
| Ensemble (Stowell et al., 2015b) | scratch init.; provided data | 78 | - |
| SoundNet (Aytar et al., 2016) | pre-training; extra data | 88 | 13M |

orthogonal to WSNet, we believe that WSNet can achieve better performance by training in a similar way as SoundNet (Aytar et al., 2016) on a large amount of unlabeled video data.

### 4.2.3 URBANSOUND8K

We report the comparison results of WSNet with state-of-the-arts on UrbanSound8k in Table 6. It is again observed that WSNet significantly reduces the model size of baseline while obtaining comparative results. Both Piczak (2015b) and Salamon & Bello (2015) use pre-computed 2D features after log-mel transformation as input. In comparison, the proposed WSNet simply takes the raw wave of recordings as input, enabling the model to be trained in an end-to-end manner.

### 4.2.4 DCASE

As evidenced in Table 7, WSNet outperforms the classification accuracy of the baseline by 1% with a $100\times$ smaller model. When using an even more compact model, *i.e.* $180\times$ smaller in model size. The classification accuracy of WSNet is only one percentage lower than the baseline (*i.e.* has only one more incorrectly classified sample), verifying the effectiveness of WSNet. Compared with SoundNet (Aytar et al., 2016) that utilizes a large number of unlabeled data during training, WSNet ($S_8C_4D_2Q_4$) that is $100\times$ smaller achieves comparable results only by using the provided data.

## 5 CONCLUSION

In this paper, we present a class of **W**eight **S**ampling networks (WSNet) which are highly compact and efficient. A novel weight sampling method is proposed to sample filters from condensed filters which are much smaller than the independently trained filters in conventional networks. The weight sampling in conducted in two dimensions of the condensed filters, *i.e.* by spatial sampling and channel sampling. Taking advantage of the overlapping property of the filters in WSNet, we propose an integral image method for efficient computation. Extensive experiments on four audio classification

datasets including MusicDet200K, ESC-50, UrbanSound8K and DCASE clearly demonstrate that WSNet can learn compact and efficient networks with competitive performance.

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

# 6 SUPPLEMENTARY MATERIAL

## 6.1 ABLATIVE STUDY OF WSNET ON ESC-50

We conduct extensive ablative study of WSNet on the public dataset ESC-50 to investigate the effects of each component of WSNet on final performance. The results are listed in Table 8.

Table 8: Studies on the effects of different settings of WSNet on the model size, computation cost (in terms of #mult-adds) and classification accuracy on ESC-50. Please refer to Table 3 for the meaning of symbols S/C/D/Q.

| WSNet's settings | conv{1-4} | | | conv5 | | | conv6 | | | conv7 | | | conv8 | | | Acc. | Model size | Mult-Adds |
|---|---|---|---|---|---|---|---|---|---|---|---|---|---|---|---|---|---|---|
| | S | C | A | S | C | A | S | C | A | S | C | A | S | C | A | | | |
| Baseline | 1 | 1 | 1 | 1 | 1 | 1 | 1 | 1 | 1 | 1 | 1 | 1 | 1 | 1 | 1 | 66.0 ± 0.2 | 13M (1×) | 2.4e8 (1×) |
| BaselineQ$_4$ | 1 | 1 | 1 | 1 | 1 | 1 | 1 | 1 | 1 | 1 | 1 | 1 | 1 | 1 | 1 | 65.7 ± 0.2 | 4× | 1× |
| S$_2$ | 2 | 1 | 1 | 2 | 1 | 1 | 2 | 1 | 1 | 2 | 1 | 1 | 2 | 1 | 1 | 66.6 ± 0.3 | 2× | 1× |
| S$_4$ | 4 | 1 | 1 | 4 | 1 | 1 | 4 | 1 | 1 | 4 | 1 | 1 | 4 | 1 | 1 | 66.3 ± 0.1 | 4× | 1.6× |
| S$_8$ | 8 | 1 | 1 | 4 | 1 | 1 | 4 | 1 | 1 | 4 | 1 | 1 | 8 | 1 | 1 | 65.2 ± 0.1 | 7× | 4.7× |
| C$_2$ | 1 | 2 | 1 | 1 | 2 | 1 | 1 | 2 | 1 | 1 | 2 | 1 | 1 | 2 | 1 | 66.8 ± 0.2 | 2× | 1× |
| C$_4$ | 1 | 4 | 1 | 1 | 4 | 1 | 1 | 4 | 1 | 1 | 4 | 1 | 1 | 4 | 1 | 66.5 ± 0.3 | 4× | 1.6× |
| C$_8$ | 1 | 8 | 1 | 1 | 4 | 1 | 1 | 4 | 1 | 1 | 4 | 1 | 1 | 8 | 1 | 65.8 ± 0.3 | 8× | 2.8× |
| S$_4$C$_4$ | 4 | 4 | 1 | 4 | 4 | 1 | 4 | 4 | 1 | 4 | 4 | 1 | 4 | 4 | 1 | 65.6 ± 0.3 | 16× | 6.3× |
| S$_8$C$_8$ | 4 | 4 | 1 | 4 | 8 | 1 | 4 | 8 | 1 | 4 | 8 | 1 | 8 | 8 | 1 | 65.2 ± 0.3 | 60× | **18.1×** |
| S$_8$C$_4$D$_2$ | 4 | 4 | 2 | 4 | 4 | 1 | 4 | 4 | 1 | 4 | 4 | 1 | 8 | 4 | 1 | **66.5 ± 0.1** | 25× | 2.3× |
| S$_8$C$_4$D$_2$Q$_4$ | 4 | 4 | 2 | 4 | 4 | 1 | 4 | 4 | 1 | 4 | 4 | 1 | 8 | 4 | 1 | 66.2 ± 0.1 | 100× | 2.3× |
| S$_8$C$_8$D$_2$ | 4 | 4 | 1 | 4 | 8 | 1 | 4 | 8 | 1 | 4 | 8 | 1 | 8 | 8 | 1 | 66.1 ± 0.0 | 45× | **2.4×** |
| S$_8$C$_8$D$_2$Q$_4$ | 4 | 4 | 2 | 4 | 8 | 1 | 4 | 8 | 1 | 4 | 8 | 1 | 8 | 8 | 1 | 65.8 ± 0.0 | **180×** | 2.4× |

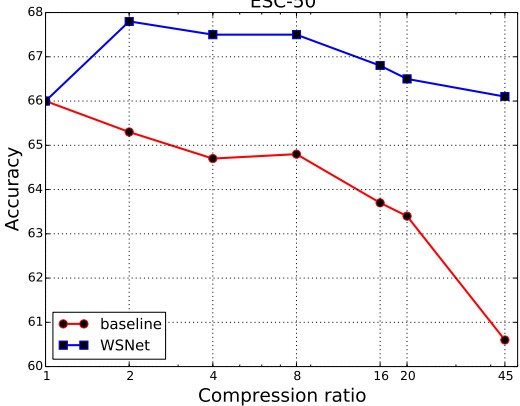
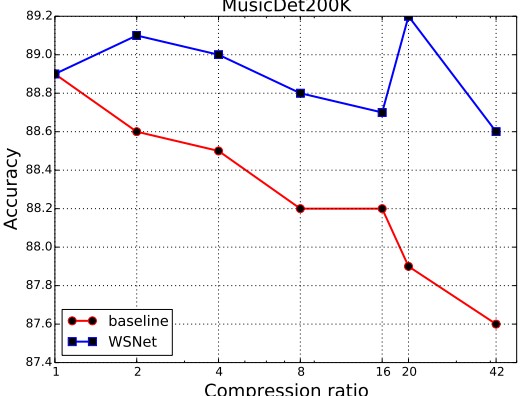

Figure 4: The accuracies of baselines and WS-Net with the same model size on ESC-50 dataset. Note the compression ratios (or compactness for WSNet) are shown in log scale.

Figure 5: The accuracies of baselines and WS-Net with the same model size on MusicDet200K dataset. Note the compression ratios (or compactness for WSNet) are shown in log scale.

## 6.2 COMPARISON BETWEEN WSNET AND BASELINES WITH REDUCED FILTERS

To further verify WSNet's capacity of learning compact models, we conduct experiments on ESC-50 and MusicDet200K to compare WSNet with baselines compressed in an intuitive way, *i.e.* reducing the number of filters in each layer. If #filters in each layer is reduced by $T$, the overall #parameters in baselines is reduced by $T^2$ (*i.e.* the compression ratio of model size is $T^2$). In Figure 4 and Figure 5, we plot how baseline accuracy varies with respect to different compression ratios and the accuracies of WSNet with the same model size of compressed baselines.

As shown in Figure 4 and Figure 5, WSNet outperforms baselines by a large magin across all compression ratios. Particularly, when the comparison ratios are large (*e.g.* 45 on ESC-50 and 42 on MusicDet200K),

Table 9: Configurations of the baseline network Chen et al. (2016) used on CIFAR10. Each convolutional layer is followed by a nonlinearity layer (*i.e.* ReLU). There are max-pooling layers (with size of 2 and stride of 2) and drop out layers following conv2, conv4 and conv5. The nonlinearity layers, max-pooling layers and dropout layers are omitted in the table for brevity. The padding strategies are all "size preserving".

| Layer | conv1 | conv2 | conv3 | conv4 | conv5 | fc1 |
|---|---|---|---|---|---|---|
| Filter sizes | 5×5 | 5×5 | 5×5 | 5×5 | 5×5 | 4096 |
| #Filters | 32 | 64 | 64 | 128 | 256 | 10 |
| Stride | 1 | 1 | 1 | 1 | 1 | 1 |
| #Params | 2K | 51K | 102K | 205K | 819K | 40K |

Table 10: Test error rates (in %) of WSNet and HashNet on MNIST. The model size is provided for the baseline. For the model size of WSNet/HashNet, we provide the ratio of the baseline's model size versus the model size of WSNet/HashNet. For the hidden fully connected layer in WSNet, we conduct weight sampling along the spatial dimension as introduced in Section 3.2.2 and the resulted WSNet has compactness of 8 and 64, respectively.

| Model | Model size | Error rate |
|---|---|---|
| baseline | 800K (1×) | 1.37 |
| HashNet Chen et al. (2015) | 8× | 1.43 |
| HashNet Chen et al. (2015) | 64× | 2.41 |
| WSNet | 8× | **1.29** |
| WSNet | 64× | **1.97** |

baselines suffer severe performance drop. In contrast, WSNet achieves comparable accuracies with full-size baselines without significant drop (66.1 versus 66.0 on ESC-50 and 88.6 versus 88.9 on MusicDet200K). This clearly demonstrates the effectiveness of weight sampling methods proposed in WSNet.

## 6.3 WSNET ON 2D CONVNETS

### 6.3.1 EXTENSION OF WSNET FROM 1D CONVNET TO 2D CONVNET

In this paper, we focus on WSNet with 1D convnets. Comprehensive experiments clearly demonstrate its advantages in learning compact and computation-efficient networks. We note that WSNet is general and can also be applied to build 2D convnets. In 2D convnets, each filter has three dimensions including two spatial dimensions (*i.e.* along X and Y directions) and one channel dimension. One straightforward extension of WSNet to 2D convnets is as follows: for spatial sampling, each filter is sampled out as a patch (with the same number of channels as in condensed filter) from condensed filter. Channel sampling remains the same as in 1D convnets, *i.e.* repeat sampling in the channel dimension of condensed filter. Following the notations for WSNet with 1D convnets (ref. to Sec. 3.1), we denote the filters in one layer as $\mathbf{K} \in \mathbb{R}^{w \times h \times M \times N}$ where $(w, h, M, N)$ denote the width and height of each filter, the number of channels and the number of filters respectively. The condensed filter $\mathbf{\Phi}$ has the shape of $(W, H, M^*)$. The relations between the shape of condensed filter and each sampled filter are:

$$W = w + (\lceil \sqrt{N} \rceil - 1)S_w$$
$$H = h + (\lceil \sqrt{N} \rceil - 1)S_h \tag{12}$$
$$M = M^* \times C$$

where $S_w$ and $S_h$ are the sampling strides along two spatial dimensions and $C$ is the compactness of WSNet along channel dimension. The compactness (please refer to Sec. 3.1 for denifinition) of WSNet along spatial dimension is $\frac{WH}{whN}$. However, such straightforward extension of WSNet to 2D convnets is not optimum due to following two reasons: (1) Compared to 1D filters, 2D filters present stronger spatial dependencies between the two spatial dimensions. Nave extension may fail to capture such dependencies. (2) It is not easy to use the integral image method for speeding up WSNet in 2D convnets as in 1D convnets. Because of above problems, we believe there are more sophisticated and effective methods for applying WSNet to 2D convnets and we would like to explore in our future work. Nevertheless, we conduct following preliminary experiments on 2D convents using above intuitive extension and verify the effectiveness of WSNet in image classification tasks (on MNIST and CIFAR10).

Table 11: Test error rates (in %) of WSNet and HashNet on CIFAR10. The model size is provided for the baseline. For the model size of WSNet/HashNet, we provide the ratio of the baseline's model size versus the model size of WSNet/HashNet. For each convolutional layer in WSNet with compactness of 16, we set its compactness along spatial/channel dimension to be 4/4, respectively. For each convolutional layer in WSNet with compactness of 64, we set its compactness along spatial/channel dimension to be 8/8, respectively. The compactness of fully connected layers in WSNet with compactness of 16 and 64 are set to be 16 and 64 respectively.

| Model | Model size | Error rate |
|---|---|---|
| baseline | 1.2M ($\times$) | 14.91 |
| HashNet Chen et al. (2016) | 16$\times$ | 21.42 |
| HashNet Chen et al. (2016) | 64$\times$ | 30.79 |
| WSNet | 16$\times$ | **17.82** |
| WSNet | 64$\times$ | **23.59** |

### 6.3.2 PRELIMINARY EXPERIMENTAL RESULTS

Since both WSNet and HashNet Chen et al. (2015; 2016) explore weights tying, we compare them on MNIST and CIFAR10. For fair comparison, we use the same baselines used in Chen et al. (2015; 2016). The baseline used for MNIST is a 3-layer fully connected network with a single hidden layer containing 1,000 hidden units. The configuration of the baseline network used for CIFAR10 is listed in Table 9. All hyperparameters used training including learning rate, momentum, drop out and so on follow Chen et al. (2015; 2016). For each dataset, we hold out 20% of the whole training samples to form a validation set. The comparison results between WSNet and HashNet on MNIST/CIFAR10 are listed in Table 10/Table 11, respectively.

As one can observe in Table 10 and Table 11, when learning networks with the same sizes, WSNet achieves significant lower error rates than HashNet on both datasets. Above results clearly demonstrate the advantages of WSNet in learning compact models.

Furthermore, we also conduct experiment on CIFAR10 with the state-of-the-art ResNet18 He et al. (2016) network as baseline. Both the network architecture and training hyperparameters follow He et al. (2016). WSNet is able to achieve 20$\times$ smaller model size with slight performance drop (0.6%). Such promising results further demonstrate the effectiveness of WSNet.

