# OpenReview forum: "WSNet: Learning Compact and Efficient Networks with Weight Sampling"
_ICLR.cc/2018/Conference — Invite to Workshop Track_

### Official Review · AnonReviewer3 · 2017-11-27
**Review of WSNet**

**Rating:** 6
**Confidence:** 4

**Review:**

In this work, the authors propose a technique to compress convolutional and fully-connected layers in a network by tying various weights in the convolutional filters: specifically within a single channel (weight sampling) and across channels (channel sampling). When combined with quantization, the proposed approach allows for large compression ratios with minimal loss in performance on various audio classification tasks. Although the results are interesting, I have a number of concerns about this work, which are listed below:

1. The idea of tying weights in the neural network in order to compress the model is not entirely new. This has been proposed previously in the context of feed-forward networks [1], and convolutional networks [2] where the choice of parameter tying is based on hash functions which ensure a random (but deterministic) mapping from a small set of “true” weights to a larger set of “virtual” weights. I think it would be more fair to compare against the HashedNet technique.

References:
[1] Wenlin Chen, James T. Wilson, Stephen Tyree, Kilian Q. Weinberger, and Yixin Chen. 2015. Compressing neural networks with the hashing trick. In Proceedings of the 32nd International Conference on International Conference on Machine Learning - Volume 37 (ICML'15), Francis Bach and David Blei (Eds.), Vol. 37. JMLR.org 2285-2294.
[2] Wenlin Chen, James Wilson, Stephen Tyree, Kilian Q. Weinberger, and Yixin Chen. 2016. Compressing Convolutional Neural Networks in the Frequency Domain. In Proceedings of the 22nd ACM SIGKDD International Conference on Knowledge Discovery and Data Mining (KDD '16). ACM, New York, NY, USA, 1475-1484. DOI: https://doi.org/10.1145/2939672.2939839

2. Given that the experiments are conducted on tasks where there isn’t a large amount of training data, one concern is that the baseline model used by the authors might be overparameterized. It would be interesting to see how performance varies as a function of number of parameters for these tasks without any “compression”, i.e., just by reducing filter sizes, for example.

3. It seems somewhat surprising that repeating the filter weights across channels as is done in the channel sharing technique yields no loss in accuracy, especially for the deeper convolutional layers. Could this perhaps be a function of the tasks that the binary “music detection” task that these models are evaluated on? Do the authors have any comments on why this doesn't hurt performance?

4. In citing relevant previous work, the authors should also include student-teacher approaches [1, 2] and distillation [3], and work by Denil et al. [4] on compression.
References:
[1] C. Bucilua, R. Caruana, and A. Niculescu-Mizil. Model compression. In Proceedings of the 12th ACM SIGKDD international conference on Knowledge discovery and data mining, pages 535–541. ACM, 2006
[2] J. Ba and R. Caruana. Do deep nets really need to be deep? In Advances in neural information processing systems, pages 2654–2662, 2014.
[3] G. Hinton, O. Vinyals, J. Dean. Distilling the Knowledge in a Neural Network, NIPS 2014 Deep Learning Workshop. 2014.
[4] M. Denil, B. Shakibi, L. Dinh, N. de Freitas, et al. Predicting parameters in deep learning. In Advances in Neural Information Processing Systems, pages 2148–2156, 2013.

5. Section 3, where the authors describe the proposed techniques is somewhat confusing to read, because of a lack of detailed mathematical explanations of the proposed techniques. This makes the paper harder to understand, in my view. Please re-write these sections in order to clearly express the parameter tying mechanism. In particular, I had the following questions:
- Are weights tied across layers i.e., are the “weight sharing” matrices shared across layers?
- There appears to be a typo in Equation 3: I believe it should be m = m* C.
- Filter augmentation/Weight quantization are applicable to all methods, including the baseline. It would therefore be interesting to examine how they affect the baseline, not just the proposed system.
- Section 3.5, on using the “Integral Image” to speed up computation was not clear to me. In particular, could the authors re-write to explain how the computation is computed efficiently with “two subtraction operations”. Could the authors also clarify the savings achieved by this technique?

6. Results are reported on the various test sets without any discussion of statistical significance. Could the authors describe whether the differences in performance on the various test sets are statistically significant?

7. On the ESC-50, UrbanSound8K, and DCASE tasks, it is a bit odd to compare against previous baselines which use different input features, use different model configurations, etc. It would be much better to use one of the previously published configurations as the baseline, and apply the proposed techniques to that configuration to examine performance. In particular, could the authors also use log-Mel filterbank energies as input features similar to (Piczak, 2015) and (Salomon and Bello, 2015), and apply the proposed techniques starting from those input features? Also, it would be useful when comparing against previously published baselines to indicate total number of independent parameters in the system in addition to accuracy numbers.

8. Minor Typographical Errors: There are a number of minor typographical/grammatical errors in the paper, some of which are listed below:
- Abstract: “Combining weight quantization ...” → “Combining with weight quantization ...”
- Sec 1: “... without sacrificing the loss of accuracy” → “... without sacrificing accuracy”
- Sec 1: “Above experimental results strongly evident the capability of WSNet …” → “Above experimental results strongly evidence the capability of WSNet …”
- Sec 2: “... deep learning based approaches has been recently proven ...” → “... deep learning based approaches have been recently proven ...”
- The work by Aytar et al., 2016 is repeated twice in the references.

---

### Official Review · AnonReviewer1 · 2017-11-27

**Rating:** 6
**Confidence:** 3

**Review:**

The paper presents a method to compress deep network by weight sampling and channel sharing.  The method combined with weight quantization provides 180x compression with a very small accuracy drop.

The method is novel  and tested on multiple audio classification datasets and results show a good compression ratio with a negligible accuracy drop.  The organization of the paper is good. However, it is a bit difficult to understand the method. Figure 1 does not help much. Channel sharing part in Figure 1 is especially confusing; it looks like the whole filter has the same weights in each channel. Also it isn’t stated in Figure and text that the weight sharing filters are learned by training.

It would be a nice addition to add number of operations that are needed by baseline method and compressed method with integral image.

Table 5: Please add network size of other networks (SoundNet and Piczak ConvNet).  For setting, SoundNet has two settings, scratch init and unlabeled video, what is that setting for WSNet and baseline?

---

### Official Review · AnonReviewer2 · 2017-11-28

**Rating:** 5
**Confidence:** 5

**Review:**

This paper presents a method for reducing the number of parameters of neural networks by sharing the set of weights in a sliding window manner, and replicating the channels, and finally by quantising weights. The paper is clearly written and results seem compelling but on a pretty restricted domain which is not well known. This could have significance if it applies more generally.

Why does it work so well? Is this just because it acts on audio and these filters are phase shifted?
What happens with 2D convnets on more established datasets and with more established baselines?
Would be interesting to get wall clock speed ups for this method?

Overall I think this paper lacks the breadth of experiments, and to really understand the significance of this work more experiments in more established domains should be performed.

Other points:
- You are missing a related citation "Speeding up Convolutional Neural Networks with Low Rank Expansions" Jaderberg et al 2014
- Eqn 2 should be m=m* x C
- Use \citep rather than \cite

---

### Decision · Program_Chairs · 2018-01-29
**ICLR 2018 Conference Acceptance Decision**

**Decision:**

Invite to Workshop Track

**Comment:**

The paper received generally positive reviews, but the reviewers also had some concerns about the evaluations.

Pros:
-- An improvement over HashNet, the model ties weights more systematically, and gets better accuracy.
Cons:
-- Tying weights to compress models already tried before.
-- Tasks are all small and/or audio related.
-- Unclear how well the results will generalize for 2D convolutions.
-- HashNet results are preliminary; comparisons with HashNet missing for audio tasks.

Given the expert reviews, I am recommending the paper to the workshop track.